# Dynamic Evolution of SARS-CoV-2 in a Patient on Chemotherapy

**DOI:** 10.3390/v15081759

**Published:** 2023-08-18

**Authors:** Weihua Huang, Changhong Yin, Kimberly P. Briley, William A. B. Dalzell, John T. Fallon

**Affiliations:** 1Department of Pathology and Laboratory Medicine, Brody School of Medicine, East Carolina University, Greenville, NC 27834, USA; yinc20@ecu.edu (C.Y.); brileyki@ecu.edu (K.P.B.); fallonj19@ecu.edu (J.T.F.); 2Department of Pediatrics, Brody School of Medicine, East Carolina University, Greenville, NC 27834, USA; dalzellw@ecu.edu

**Keywords:** SARS-CoV-2, COVID-19, deletion, evolution, chemotherapy

## Abstract

Severe acute respiratory syndrome coronavirus 2 (SARS-CoV-2) has evolved significantly during the pandemic and resulted in daunting numbers of genomic sequences. Tracking SARS-CoV-2 evolution during persistent cases could provide insight into the origins and dynamics of new variants. We report here a case of B-cell acute lymphocytic leukemia on chemotherapy with infection of SARS-CoV-2 for more than two months. Genomic surveillance of his serial SARS-CoV-2-positive specimens revealed two unprecedented large deletions, Δ15–26 and Δ138–145, in the viral spike protein N-terminal domain (NTD) and demonstrated their dynamic shifts in generating these new variants. Located at antigenic supersites, these large deletions are anticipated to dramatically change the spike protein NTD in three-dimensional protein structure prediction, which may lead to immune escape but reduce their viral transmissibility. In summary, we present here a new viral evolutionary trajectory in a patient on chemotherapy.

## 1. Introduction

Severe acute respiratory syndrome coronavirus 2 (SARS-CoV-2) is a positive-sense single-stranded RNA virus causing the severe disease known as coronavirus disease 2019 (COVID-19). Since its emergence at the end of 2019, this pathogen has led to a prolonged pandemic and caused millions of deaths and substantial morbidity worldwide; meanwhile, this virus has become the most sequenced pathogen in the world. From these unparalleled amounts of genomic data, we have witnessed an unprecedented dynamic evolution of SARS-CoV-2. Over the course of COVID-19 waves, SARS-CoV-2 has evolved variants of concern (VoCs) from Alpha, Delta to Omicron, according to the World Health Organization assignment; and Omicron has recently evolved from BA.1/BA.2, BA.4/BA.5 and BQ.1 to XBB, based upon major viral lineages. Notably, Omicron variants have a much higher capacity to cause reinfections and contain far more mutations in spike proteins than any variants previously reported, which led to widespread escape from neutralizing antibody responses [1,2,3].

Viral evolution is a complex process, as viruses replicate and evolve within individuals (within-host), but they must also successfully transmit person to person (between-host), resulting in evolution at a different scale [4]. Since chronic infection allows the virus to acquire many evolutionary changes, it has been hypothesized that the emergence of a novel VoCs most likely arises in immunocompromised patients with prolonged infection [5,6]. In support of this hypothesis, many site mutations and/or small indels in the viral spike protein have been identified within-host in cases of chronic SARS-CoV-2 infection, linking to either antibody resistance or higher rates of viral transmission [7,8,9,10]. We report here the identification of two unprecedented large deletions in the SARS-CoV-2 spike protein in a patient on chemotherapy and their dynamic shift to new variants during his prolonged viral infection.

## 2. Results

A 17-year-old male was diagnosed with B-cell acute lymphocytic leukemia (B-ALL) on 7 February 2022. Two weeks later, he started on a month of induction chemotherapy with intrathecal administration of high-dose methotrexate, vincristine, daunorubicin and pegaspargase via a lumbar puncture. The followed consolidation chemotherapy with high-dose methotrexate continued for about 5 months, until the patient had leukopenia as diagnosed on 26 August 2022. Experiencing fever, dizziness, headache and sore throat, the patient tested COVID-19-positive on 6 September 2022 (marked as Day 0). Nevertheless, to avoid any B-ALL treatment delay he started on his maintenance chemotherapy on 9 September 2022. In subsequent monitoring of his SARS-CoV-2 infection, at least five specimens collected from his nasopharynx (marked as Day 15, 38, 50, 60 and 63, respectively) consecutively tested positive. His SARS-CoV-2 infection persisted for more than two months, during which his maintenance chemotherapy continued. Notably, the patient received a single dose of Pfizer-BioNtech mRNA vaccine in September 2021, based on his electronic medical record. Although his immunoglobin (IgG) antibody against COVID-19 spike was detected to be rising in May 2021, his COVID-19 PCR tests before that were all negative, while his influenza A PCR test was positive on 20 May 2021. The schemed timeline regarding the patient’s B-ALL diagnosis and subsequent chemotherapy treatments, as well as his SARS-CoV-2 vaccination and infection monitoring, is summarized and demonstrated in Figure 1.

**Figure 1 viruses-15-01759-f001:**
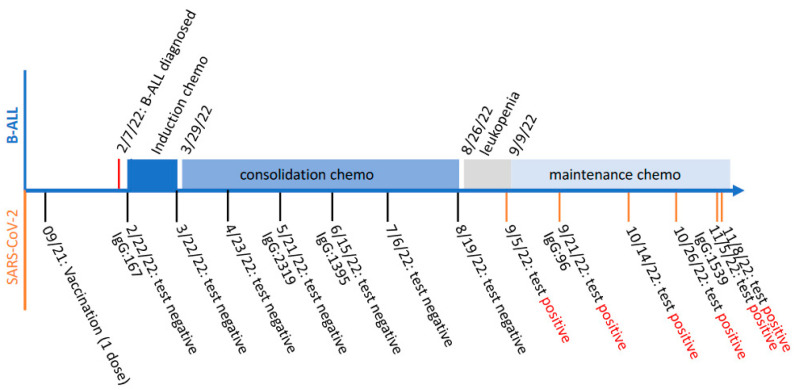
Schemed timeline of the patient’s B-cell acute lymphocytic leukemia (B-ALL) diagnosis and subsequent chemotherapy treatment (top), as well as SARS-CoV-2 vaccination and infection monitoring (bottom). The levels of immunoglobin (IgG) antibody against COVID-19 spike are shown as antibody unit per mL (AU/mL).

**Figure 2 viruses-15-01759-f002:**
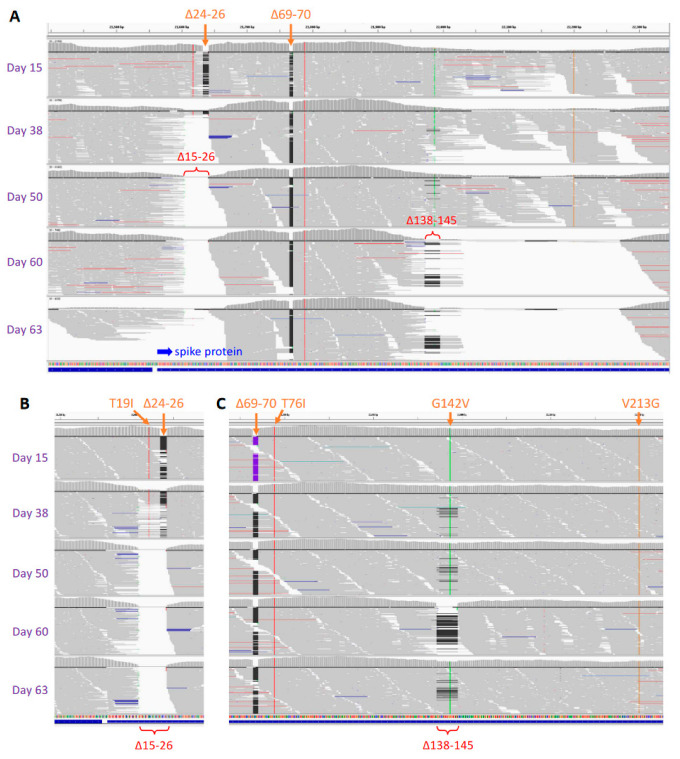
Large deletions in SARS-CoV-2 spike identified via whole-genome sequencing (**A**) and validated with target resequencing (**B**,**C**). Due to Δ138–145 deletion, genome sequencing based on multiplex PCR amplification was incomplete for specimens at and after Day 38, especially those at Days 60 and 63 (**A**), which was compensated for with subsequent target resequencing validation (**C**). Integrative Genomics Viewer (IGV) was used for visualization of sequence reads alignment and mapping to the SARS-CoV-2 genome reference NC_045512 from Wuhan-Hu-1 isolate. Site mutations and small deletions in lineage BA.5.5 are shown in orange on the top. Large deletions, Δ15–26 and Δ138–145, are shown in red. Blue arrow indicates the N terminal of spike protein.

Whole-genome sequencing of SARS-CoV-2 was conducted on his serial COVID-19-positive specimens, using the improved v3 primer set [11] from ARTIC Network, a consortium dedicated to global viral epidemiology efforts (https://artic.network (accessed on 26 July 2023)). The results of genomic sequencing intriguingly revealed two novel large deletions (Figure 2 and Table 1): Δ15–26, a 36-nucleotide (nt) deletion; Δ139–145 or Δ138–145, a 21- or 24-nt deletion; and a site mutation R346I in the viral spike protein. All these deletions and the mutation were validated via the subsequent target resequencing. Frequency analysis of viral population in these specimens further revealed their dynamic shifts in generating novel SARS-CoV-2 variants. As detected at Day 15, the primary lineage initially belonged to Omicron BA.5.5, with both Δ24–26 and Δ69–70 deletions in the spike N-terminal domain (NTD). While Δ69–70 deletion remained unchanged, Δ24–26 deletion was found enlarged to Δ15–26, shown 57% in the viral population at Day 38 and 100% thereafter. Deletions Δ139–145 or Δ138–145 were caught in low frequency at Day 38 and Day 50 (≤10%, respectively), but Δ138–145 became dominant (100%) at Day 60, suggesting viruses with Δ138–145 were more competitive in viral amplification and selection than those with Δ139–145. However, a subsequently observed reduction in Δ138–145, 24% in the viral population at Day 63 suggested viruses with Δ138–145 were not prolonged either. In a comparison, site mutation R346I in the spike receptor bind domain (RBD) was consistently detected at and after Day 38. Combining Figure 2 and Table 1, we reveal here a rapid and dynamic evolution of SARS-CoV-2 in the patient on maintenance chemotherapy.

Located in the NTD, both Δ15–26 and Δ138–145 deletions are novel and large, with N-glycosylation sites Asn17 deleted and Asn122 and Asn149 possibly jeopardized. These deletions made the NTD more compact, based on our prediction of its protein three-dimensional structures using a language-model-based ESM Atlas [12] (Figure 3). Compared to that in lineage B of the Wuhan-Hu-1 isolate, deletions Δ24–26 and Δ69–70 in BA.5.5 significantly altered the NTD’s structure. In contrast, the unprecedented large deletions, Δ15–26 and Δ138–145, mediated conformational changes not only at local NTD surface loops but also in their neighbors, and reorganized the NTD’s structure more dramatically. As demonstrated in Figure 3, Δ138–145 might change some of the β-sheets in BA.5.5 into α-helixes.

## 3. Discussion

We reveal here the dynamic in vivo evolution of SARS-CoV-2 in a patient on chemotherapy. The newly identified large deletions in the viral spike protein are unprecedented. However, their mutation sites have been described previously. Compared to Δ15–26, deletion Δ24–26 at NTD surface loop N1 was common in Omicron BA.2, BA.4 and BA.5, and was found to reduce viral infectivity [13]. Compared to Δ138–145, deletion Δ144 was present in Alpha B.1.1.7 and deletion Δ143–145 was present in Omicron BA.1, the latter leading to a reconfiguration of NTD surface loop N3 from a hairpin fold to a loose loop and causing immune evasion [3,14]. Additionally, deletion Δ69–70 was found in Alpha B.1.1.7 and Omicron BA.1, BA.4 and BA.5, which triggered conformational changes in NTD surface loop N2 and altered the antigenic “supersite” [3,14]. Residing at the RBD, R346 had multiple variants, including R346K in Mu B.1.621, R346S in BA.4.7, R346I in BA.5.9 and R346T in BF.7, which featured substantial neutralization resistance but preserved the viral activity of binding to angiotensin converting enzyme 2 (ACE2) [6,15,16], a receptor for SARS-CoV-2. Since N1-N3 surface loops are antigenic “supersites” for NTD-directed neutralizing antibodies [1,2,3], it is anticipated that our newly identified variants would cause dramatic changes in spike protein structure as well as N-glycosylation, though compared to cryogenic electron microscopy analysis of the whole spike protein, our local NTD protein structure prediction is very limited. It is also anticipated that the dramatically reorganized NTD would evade antibody recognition, leading to vaccine breakthrough, just like Δ143–145. However, to the best of our knowledge, these new variants do not seem to have spread to anyone, suggesting their low infectivity (e.g., Δ15–26, like Δ24–26) or low stability (e.g., Δ138–145, disappearing at Day 63), leading to their low transmissibility.

We document here a single case of a B-ALL patient with a prolonged COVID-19 viral replication for more than two months. Due to the pandemic, the patient has been monitored closely for SARS-CoV-2 infection during his whole chemotherapy treatment. With his infection onset at leukopenia status, the subsequently prolonged viral shedding is unusual, likely due to immunosuppression after completion of induction and consolidation chemotherapy and during maintenance chemotherapy. Were the viral mutations in this patient driven by drug treatments in chemotherapy? Possibly, since these deletions were rare and large. However, other factors might have contributed significantly as well, such as his potential natural SARS-CoV-2 infection in May 2021 with COVID-19 IgG rising (though negative at COVID-19 PCR tests), subsequent COVID-19 vaccination in September 2021 and waning immunity during chemotherapy. These combined selection pressures might have led to the viral evolution. Nevertheless, the question remains to be further addressed in future more observations and/or experiments.

Using genomic sequencing, we characterized here the viral within-host evolution in response to selection pressures in an individual on chemotherapy. This unique tracking case provides additional evidence for us to understand the diversity, evolution and dynamics of SARS-CoV-2.

## 4. Materials and Methods

**Clinical SARS-CoV-2 tests:** Nasopharyngeal swab samples were collected from the patient for SARS-CoV-2 rapid RT-PCR test. Two main test platforms were used: the ePLEX Respiratory Panel 2 (GenMark Diagnostics, Carlsbad, CA, USA) targeted two unique regions of the nucleocapsid (N) gene; and the cobas SARS-CoV-2 Test (Roche Molecular Diagnostics, Pleasanton, CA, USA) targeted dual genes, open reading frame 1a/b (ORF1ab) and envelop (E), with an automated cobas 8800 instrument (Roche Molecular Diagnostics). Blood samples were collected from the patient for serological IgG test against SARS-CoV-2 nucleocapsid protein. A two-step qualitative chemiluminescent microparticle immunoassay (CMIA) was performed on an automated Architect system (Abbott Diagnostics, Des Plaines, IL, USA). All the tests were approved by the federal agency the Food and Drug Administration (FDA) under Emergency Use Authorization (EUA) and performed in CLIA-certified clinical laboratories.

**Next-generation sequencing (NGS):** RNA was extracted from serial COVID-19-positive nasopharyngeal swab samples of the patient using the Direct-zol RNA Microprep Kit (Zymo Research, Irvine, CA, USA). SARS-CoV-2 whole-genome sequencing was performed using the improved ARTIC Network’s v3 primer set [11] and multiplex PCR amplification (dx.doi.org/10.17504/protocols.io.bibtkann (accessed on 26 July 2023)), followed by the Nextera XT Library Prep kit (Illumina, San Diego, CA, USA) according to the manufacturer’s instructions. DNA quantity was determined via the AccuClear Nano dsDNA Assay and SpectraMax iD3 fluorometer (Molecular Devices, San Jose, CA, USA). Library quality was verified via the 4200 TapeStation (Agilent, Santa Clara, CA, USA). NGS was performed with paired-end sequencing (75 × 2 cycles) in the MiSeq (Illumina) platform. Due to Δ138–145 deletion, primer 73Left was not in full function for specimens at and after Day 38, especially those specimens at Day 60 and Day 63, leaving a gap in the genome sequence assembly. Validation of deletions was conducted using PCR amplification with two primer sets, 71Left and 71Right for Δ15–26 determination and 72Left and 73Right for Δ138–145 determination, followed by Illumina target resequencing at a higher depth using the Nextera XT Library Prep method. Primers 71Left, 71Right, 72Left, 73Left and 73Right were the same as in the improved ARTIC Networks’ v3 primer set.

**Bioinformatics analysis:** After removal of sequencing adaptors using trimmomatic (v0.39), raw sequence reads were aligned and mapped to the SARS-CoV-2 reference genome (Wuhan-Hu-1 strain, GenBank accession no. NC_045512) for whole-genome assembly, and variants were identified using the snippy algorithm (https://github.com/tseemann/snippy (accessed on 26 July 2023)). The lineage of the assembled nearly complete viral genome was identified using the Pangolin lineage assigner and PANGO lineage database [17], both of which were updated in a timely manner. Raw sequence data, including the validation sequencing, were submitted to the SRA database (accession numbers: SRR22465584–SRR22465598). The main genome sequence for each specimen was submitted to the GISAID database (accession numbers: EPI_ISL_15966605–EPI_ISL_15966608). Notably, due to the low viral load, the assembled genome for the specimen at Day 63 had many gaps and was not complete enough for GISAID submission. Deletions and mutations were manually inspected and confirmed under the Integrative Genomics Viewer [18] (IGV). All mutations were relative to the SARS-CoV-2 Wuhan-Hu-1 genome sequence NC_045512 and its viral spike protein sequence YP_009724390.1. For target resequencing validation, the BWA aligner [19] (v0.7.17-r1188), instead of snippy, was used for sequence read alignment and mapping. The mutation frequency for each deletion was estimated from the target sequencing under IGV manual inspection.

**Protein structure prediction:** the three-dimensional protein structures of spike NTD, relative to residues 1–360 of YP_009724390.1 with partial RBD included (residues 331–360), were predicted using the “Fold Sequence” function in a language-model-based ESM Atlas (https://esmatlas.com/ (accessed on 26 July 2023)).

## Figures and Tables

**Figure 3 viruses-15-01759-f003:**
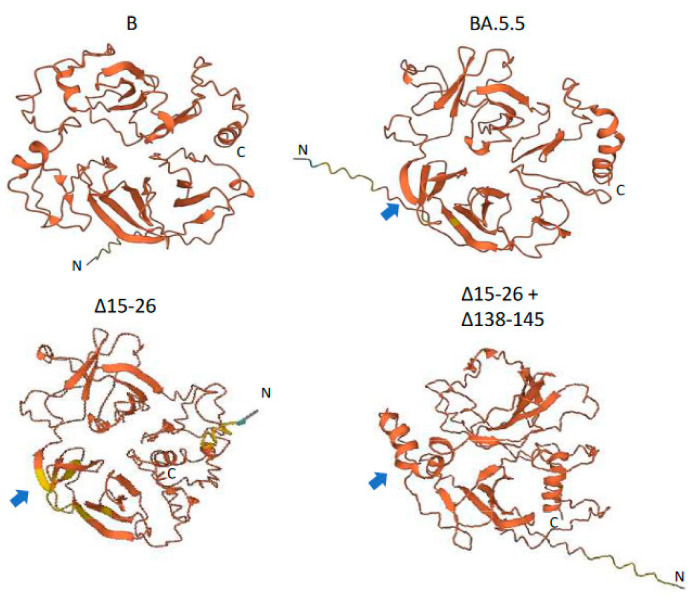
Predicted protein structure of SARS-CoV-2 spike protein N-terminal domain (NTD). NTD protein structure was predicted using a language-model-based ESM Atlas [12] with residues 1–360 of spike protein relative to YP_009724390.1 from lineage B. Notably, residues 331–360, including the C-terminal α-helix, belong to the spike receptor binding domain (RBD). Blue arrows demonstrate the structural change from β-sheets in lineage BA.5.5 to α-helixes due to Δ138–145 deletion.

**Table 1 viruses-15-01759-t001:** Dynamic spike protein mutational shifts identified via SARS-CoV-2 whole-genome sequencing, validated and quantified with target resequencing. Genome reference NC_045512 and spike protein reference YP_009724390.1 were used. Serial COVID-19-positive nasopharyngeal swab specimens were collected from the same patient receiving chemotherapy.

Specimen	Collection Date	Deletion Site 1		Deletion Site 2	Deletion Site 3		Site Mutation	Lineage
		Δ24–26	Δ15–26	Δ69–70	Δ139–145	Δ138–145		
Day 15	21 September2022	100%	-	100%	-	-	-	BA.5.5
Day 38	14 October 2022	43%	57%	100%	10%	4%	R346I	novel
Day 50	26 October 2022	-	100%	100%	9%	6%	R346I	
Day 60	5 November 2022	-	100%	100%	-	100%	R346I	
Day 63	8 November 2022	-	100%	100%	-	24%	R346I	

## Data Availability

Raw sequence data were submitted to the SRA database (accession numbers: SRR22465584–SRR22465598). The assembled genome sequences were submitted to the GISAID database (accession numbers: EPI_ISL_15966605–EPI_ISL_15966608).

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
