# Peer review of "Dynamic Evolution of SARS-CoV-2 in a Patient on Chemotherapy"

_viruses, 2023, doi:10.3390/v15081759_

Round 1

Reviewer 1 Report

“Dynamic Evolution of SARS-CoV-2 in a Patient on Chemotherapy” describes  a case of a patient with B-cell acute lymphocytic leukemia on chemotherapy, with a COVID-19 infection longer than two months. Genomic surveillance of his serial COVID-19-positive specimens revealed two new large deletions, not detected before. The paper is very interesting and provides new useful information about the persistent infection of Sars-CoV-2 in immunocompromised patients and the evolutionary dynamic of Sars-CoV-2 during prolonged infections, that can lead to new variants, and  consequently to immune escape. Of course these results should be confirmed by other similar studies on this topic.

The paper is quite clear and well described, I just would add some information about the clinical case and its description to the paragraph  “Materials and Methods”, that can be moved from the “Results”.

Minor revisions:

In the text I see a lot of double spaces at the beginning of the sentences.

Please differentiate the characters of the captions from those of the text. They are the same and this is confusing, since it’s not clear where the caption ends.

Line 101: Correct “in complete” with “incomplete”.

Author Response

We greatly appreciate the reviewer’s positive comments and constructive suggestions.

The paper is quite clear and well described, I just would add some information about the clinical case and its description to the paragraph “Materials and Methods”, that can be moved from the “Results”.

We do appreciate the reviewer’s concern. However, since the “Materials and Methods” is the last section of the manuscript, we think it will be more convenient for readers to be aware of the clinical case at the very beginning and get the full story smoothly.

Minor revisions:

In the text I see a lot of double spaces at the beginning of the sentences.

Yes, we used double spaces between the sentences through the whole manuscript. We leave this concern to the journal editors.

Please differentiate the characters of the captions from those of the text. They are the same and this is confusing, since it’s not clear where the caption ends.

We added a line between the caption and the text, and also would like to leave this to the journal editors.

Line 101: Correct “in complete” with “incomplete”.

Corrected in the revision. Thanks.

Reviewer 2 Report

I have no critical comments. In spite of this, my positive impression is based on the careful reading of the manuscript. 

Author Response

We greatly appreciate the reviewer's time and positive commends. Many thanks,

Reviewer 3 Report

In this work, Huang and colleagues present a clinical case a young man with B-ALL with infection of SARS-CoV-2 for more than two months and the genomic evolution of the virus along time in the patient. The authors found two unprecedented large deletions in the viral spike protein NTD and a point mutation. This brief report is well presented and written; the data are clearly described and the conclusions well balanced. I think the results of this report could be important for the scientific community in order not to underestimate the risk of the intra-host evolution of SRAS-CoV-2.

Given all these considerations, I have just few minor points:

1.       In the Materials and Methods section, please expand a bit more the procedure for the creation of the in silico structure of the viral spike protein NTD

2.       In the Materials and Methods section, please add the procedures used for the RNA isolation and molecular testing of the patient samples during time, plus the serological analyses for the antibodies titer.

3.       If possible, please try to ameliorate the quality of the Figure 2 as it is a bit difficult to read

Author Response

We greatly appreciate the reviewer’s positive comments and constructive suggestions.

  1. In the Materials and Methods section, please expand a bit more the procedure for the creation of the in silico structure of the viral spike protein NTD

We would love to add more as the reviewer suggested. However, the ESM Atlas is quite easy to use. What we need to do is just input the protein sequence on the website and change the angle to view the 3D-structure after it shows up.

  1. In the Materials and Methods section, please add the procedures used for the RNA isolation and molecular testing of the patient samples during time, plus the serological analyses for the antibodies titer.

Yes, thanks for the suggestion. We have added these details in the revision.

  1. If possible, please try to ameliorate the quality of the Figure 2 as it is a bit difficult to read

      We adjusted the size of figures in the revision and would like to leave this to the journal editors for further adjustment.

Reviewer 4 Report

In the paper by Huang et al., the authors analyse a SARS-CoV-2 persistently infected patient undergoing chemotherapy. The report is well written and the bioinformatic analyses of the intra-host viral evolution are largely sound. However, at some instances the results are likely over interpreted and/or lack some basic analyses. Please find in the following my evaluation:

Major comments:

- The novelty of the two large deletions (delta15-26, delta138-145) was not shown. For example, the R package "outbreak.info R Client" (outbreak-info.github.io) can access mutational data from GISAID and would allow to analyse if the described mutations have been previously detected and in which variants.

- It is unclear, why the authors used the language model to predict the SARS-CoV-2 secondary structure and why they limit their prediction to the NTD. An AlphaFold2 prediction and/or a prediction of the full-length spike protein would likely be beneficial for interpreting the effect of their detected deletions for the NTD and the full-length spike.

- Table 1. The sharp increase of delta138-145 deletion between day 50 and 60 followed by the rapid decline at day 63 is highly puzzling, as this counter selection is rather quick. Can the authors exclude that these shifts were not caused by low viral RNA input and subsequent low coverage? Were the frequencies calculated from the targeted validation or from the full-length sequencing? Please clarify!

- Discussion line 17: Please rephrase your statement that the NTD mutations would cause vaccine breakthroughs. Although neutralizing ab are generated against the NTD, the main antigenic target structure during infection is the RBM. It was not shown/predicted here that these deletions cause conformational changes for the full length spike. 

Minor comments:

- Figure 1: IgG levels is missing the unit (BAU/ml ?). Moreover, information regarding the test used to determine antibody titers is missing in the Material and Methods.

- Was the patient treated with any antivirals or monoclonal antibodies? Mutations at S position 346 were also associated with Evusheld escape and are now quite common. Therefore, the de novo development of this mutation is intriguing and could hint to a specific selection pressure (e.g. monoclonals).

- ESM atlas has multiple language models. Please specify which was used in your analysis.  

- Line 41: change species to pathogen

- Line 52: The hypothesis that a VoC arise in immunocompromised patients is not valid for all VoC. As far as I am aware this was mainly hypothesized for the Alpha and Omicron variant. Please consider rephrasing.

- Have the authors tried to isolate the virus and is it viable in cell culture? 

- Please give a hypothesis why the patient was able to clear the virus. Was this due to increasing antibody titers? Indicating changes in the Ct value could give a hint in that regard.

- Please indicate in the Material and Methods the target gene for qPCR and the test used.

 Discussion line 25/26: The authors hypothesise that viral evolution was driven by chemotherapy drugs. Has this been observed before and/or can the authors provide a molecular explanations how this could have happened? It is contra-intuitive that a non-spike specific selection pressure would cause this. Do the authors find other strange mutations/deletions outside of the spike gene that could collaborate this hypothesis?

Well written. No comments.

Author Response

We greatly appreciate the reviewer’s positive comments and constructive suggestions.

In the paper by Huang et al., the authors analyse a SARS-CoV-2 persistently infected patient undergoing chemotherapy. The report is well written and the bioinformatic analyses of the intra-host viral evolution are largely sound. However, at some instances the results are likely over interpreted and/or lack some basic analyses. Please find in the following my evaluation:

Major comments:

- The novelty of the two large deletions (delta15-26, delta138-145) was not shown. For example, the R package "outbreak.info R Client" (outbreak-info.github.io) can access mutational data from GISAID and would allow to analyse if the described mutations have been previously detected and in which variants.

We appreciate the reviewer’s information and concern. As far as we are aware of, the package is strong in tracking the known lineages with known variants, but weak in tracking deletions, especially large deletions. This is probably due to 1) a large number of incomplete genome sequences (with N and -) were included in the GISAID database; and 2) genome sequences with too many gaps (low quality) were excluded in the GISAID database. In our case, the two large deletions made the genome sequences assembly in low quality, not being submitted to the GISAID as we mentioned in the manuscript. Nevertheless, to the best of our knowledge, these large deletions have not been reported yet, and therefore, we used the term “unprecedented” in the manuscript.

- It is unclear, why the authors used the language model to predict the SARS-CoV-2 secondary structure and why they limit their prediction to the NTD. An AlphaFold2 prediction and/or a prediction of the full-length spike protein would likely be beneficial for interpreting the effect of their detected deletions for the NTD and the full-length spike.

Currently, the ESM algorithm has a size limitation in predict 3-D structure of a protein, not larger than 500-aa. Spike is a large protein (1273-aa) with multiple domains, The N-terminal NTD and RBD domains mainly are separated from the C terminal FP, HR1, HR2 TM and CT domain. We agree with the reviewer that the structural prediction of the whole Spike will be more beneficial and AlphaFold2 is more powerful in protein 3D-structure prediction. However, we believe our prediction with ESM in the local region is good enough to serve the purpose, to show the dramatic change of 3-D structure locally caused by the large deletions.

- Table 1. The sharp increase of delta138-145 deletion between day 50 and 60 followed by the rapid decline at day 63 is highly puzzling, as this counter selection is rather quick. Can the authors exclude that these shifts were not caused by low viral RNA input and subsequent low coverage? Were the frequencies calculated from the targeted validation or from the full-length sequencing? Please clarify!

These quantification results were from the PCR-sequencing validation, not from whole genome amplification-sequencing. As we mentioned in our manuscript, the large deletions made large gaps in the genome sequences, from which the quantification was impossible or incorrect. Therefore, we conducted PCR-sequencing validation with high coverage, the results from which had less bias from low viral RNA input. We have revised the manuscript to make it clearer, in Table 1 and in the Materials and Methods section.

- Discussion line 17: Please rephrase your statement that the NTD mutations would cause vaccine breakthroughs. Although neutralizing ab are generated against the NTD, the main antigenic target structure during infection is the RBM. It was not shown/predicted here that these deletions cause conformational changes for the full length spike. 

Our statement of anticipation was mainly based on previous studies on delta143-145 deletion, which caused immune evasion, as we cited in Discussion lines 6-8. Our deletion delta138-145 was at the same site as the delta143-145, but much larger. Though our 3-D structural prediction was based on local domain, not the whole spike protein, we believe our anticipation was very reasonable.

Minor comments:

- Figure 1: IgG levels is missing the unit (BAU/ml ?). Moreover, information regarding the test used to determine antibody titers is missing in the Material and Methods.

Thanks to the reviewer for the suggestion. In the revision, we have added the unit to the Figure 1 caption and added a paragraph of clinical tests in the Materials and Methods section.

- Was the patient treated with any antivirals or monoclonal antibodies? Mutations at S position 346 were also associated with Evusheld escape and are now quite common. Therefore, the de novo development of this mutation is intriguing and could hint to a specific selection pressure (e.g. monoclonals).

No, as far as we checked up his medical record, he was not treated with any antiviral or monoclonal antibody. We agree with the reviewer that multiple mutations on S346 are intriguing, worthy of further investigation.

- ESM atlas has multiple language models. Please specify which was used in your analysis.  

We only used the “Fold Sequence” function in the ESM Atlas. We did not get into the detail of which language model was used behind the function.

- Line 41: change species to pathogen

Changed in the revision as the reviewer suggested.

- Line 52: The hypothesis that a VoC arise in immunocompromised patients is not valid for all VoC. As far as I am aware this was mainly hypothesized for the Alpha and Omicron variant. Please consider rephrasing.

We believe the authors of hypothesis were meant to make the hypothesis universal, not only to some of variants or VoC. The more attentions on VoC were mainly because they were more deleterious to human health, and therefore these VoCs received more study.

- Have the authors tried to isolate the virus and is it viable in cell culture? 

No, we did not. Though we would have loved to, we did not have expertise to do so.

- Please give a hypothesis why the patient was able to clear the virus. Was this due to increasing antibody titers? Indicating changes in the Ct value could give a hint in that regard.

Due to chemotherapy, the patient was immunocompromised. The immune response to the virus was thus slow and diminished, compared to normal people, leading to the prolonged infection. The later increasing antibody indicated that the patient’s immune response was luckily coming back, fighting back against the virus. Accordingly, the virus titer was decreased, probably and partially due to their low viral transmissibility. During the whole process, many factors would have some influences as we discussed. But, it is hard for us to say in certain without any supporting evidence.

- Please indicate in the Material and Methods the target gene for qPCR and the test used.

Thanks for the reviewer’s suggestion. We have added in the revision a paragraph on clinical tests including the PCR test and its target genes.

 Discussion line 25/26: The authors hypothesise that viral evolution was driven by chemotherapy drugs. Has this been observed before and/or can the authors provide a molecular explanations how this could have happened? It is contra-intuitive that a non-spike specific selection pressure would cause this. Do the authors find other strange mutations/deletions outside of the spike gene that could collaborate this hypothesis?

Some of chemotherapy drugs have effects on interaction with DNA by intercalation, e.g., daunorubicin; combining with enzyme to disrupt DNA/RNA synthesis, e.g., mercaptopurine; or converting to a chemical crosslinking DNA, e.g., cyclophosphamide. These drugs may also have some influences on viral evolution, though we have little supporting evidence. Since this is the first time observing such large deletions, to the best of our knowledge, it is reasonable for us to assume these deletions somehow related to chemotherapy drugs (a hypothesis). However, more evidence is needed as we mentioned in the manuscript. Notably, no additional large deletion was found in the virus other than the spike protein.  

Round 2

Reviewer 4 Report

All comments were adressed. No further suggestions.

No comments.